# Prediction of Prognosis in Pancreatic Cancer According to Methionyl-tRNA Synthetase 1 Expression as Determined by Immunohistochemical Staining

**DOI:** 10.3390/cancers15225413

**Published:** 2023-11-14

**Authors:** Sung Ill Jang, Ji Hae Nahm, See Young Lee, Jae Hee Cho, Min-Young Do, Joon Seong Park, Hye Sun Lee, Juyeon Yang, Jiwon Kong, Seunghwan Jung, Sunghoon Kim, Dong Ki Lee

**Affiliations:** 1Department of Internal Medicine, Gangnam Severance Hospital, Yonsei University College of Medicine, Seoul 06273, Republic of Korea; aerojsi@yuhs.ac (S.I.J.); seeyoung87@yuhs.ac (S.Y.L.); jhcho9328@yuhs.ac (J.H.C.); decays@naver.com (M.-Y.D.); 2Department of Pathology, Gangnam Severance Hospital, Yonsei University College of Medicine, Seoul 06273, Republic of Korea; nam2169@yuhs.ac; 3Department of Surgery, Gangnam Severance Hospital, Yonsei University College of Medicine, Seoul 06273, Republic of Korea; jspark330@yuhs.ac; 4Biostatistics Collaboration Unit, Yonsei University College of Medicine, Seoul 06273, Republic of Korea; hslee1@yuhs.ac (H.S.L.); ju1003yeon@yuhs.ac (J.Y.); 5Institute for Artificial Intelligence and Biomedical Research, Medicinal Bioconvergence Research Center, Yonsei University, Incheon 21983, Republic of Korea; jkeykong@target.re.kr (J.K.); fersen@target.re.kr (S.J.); sunghoonkim@yonsei.ac.kr (S.K.); 6Yonsei Institute of Pharmaceutical Sciences, College of Pharmacy, Yonsei University, Incheon 21983, Republic of Korea

**Keywords:** methionyl-tRNA synthetase 1, pancreatic ductal adenocarcinoma, prognostic marker

## Abstract

**Simple Summary:**

Pancreatic ductal adenocarcinoma (PDAC) is the fourth leading cause of cancer-related deaths worldwide, but effective prognostic markers are lacking. Methionyl-tRNA synthetase 1 (MARS1), a critical enzyme in translation initiation that transfers Met to the initiator tRNA, has been implicated in cancer development and progression. MARS1 expression was significantly increased in PDAC versus normal pancreatic duct tissues. Additionally, high expression of MARS1 was associated with a poor prognosis in patients with PDAC. Our findings suggest that MARS1 is involved in pancreatic carcinogenesis and has potential as a novel prognostic marker for PDAC.

**Abstract:**

The serum level of CA 19-9 is a prognostic marker for pancreatic ductal adenocarcinoma (PDAC). We evaluated the ability of the expression level of methionyl-tRNA synthetase 1 (MARS1)—which facilitates cancer growth by modulating protein synthesis and the cell cycle—to predict the prognosis of PDAC. Immunohistochemical (IHC) staining was performed on pancreatic specimens obtained from patients with PDAC who were undergoing surgery. High MARS1 expression was defined as equal to, or greater than, that in normal acinar cells. Low MARS1 expression was defined as weaker than in normal acinar cells, and stronger than in the pancreatic duct epithelium. Univariate and multivariate analyses were performed on other factors related to prognosis. Among 137 PDAC patients, no significant differences in baseline characteristics were found between those with high (n = 82) and low (n = 55) MARS1 expression. The median overall survival time of patients with high MARS1 expression was shorter than that of those with low expression (15.2 versus 17.2 months, log-rank test *p* = 0.044). The median disease-free survival (DFS) was not significantly different between the two groups. However, the DFS was shorter in patients with high than in those with low MARS1 expression (8.9 versus 11.2 months, log-rank test *p* = 0.067). In a multivariate analysis, lymph node metastasis and high MARS1 expression were associated with a poor prognosis of PDAC. Elevated MARS1 expression detected by IHC staining is associated with a poor prognosis of PDAC, suggesting that MARS1 has potential as a prognostic marker.

## 1. Introduction

Pancreatic ductal adenocarcinoma (PDAC) has the worst prognosis of the major malignancies, with a 5-year survival rate of 6% [1]. Resectable PDAC is first treated surgically, followed by adjuvant chemotherapy [2,3,4]. For borderline resectable PDAC, neo-adjuvant chemotherapy is followed by surgery or continuing chemotherapy; palliative chemotherapy is performed for unresectable PDAC. Because of the difficulty of early diagnosis, only 10 to 20% of pancreatic cancers can be surgically resected with curative intent at the time of diagnosis [2]. In unresectable pancreatic cancer, tumor factors (such as the carbohydrate antigen [CA19–9] level and tumor stage), host factors (including performance status), the serum C-reactive protein level, and the neutrophil–lymphocyte ratio are associated with survival [5,6,7]. However, prediction of prognosis is problematic. Therefore, the discovery of novel tumor biomarkers would enhance the prevention, diagnosis, prognosis, and targeted therapy of PDAC.

Human cytoplasmic methionyl-tRNA synthetase 1 (MARS1) consists of 900 amino acids [8] and is a component of the multi-tRNA synthetase complex [9]. MARS1 is an aminoacyl-tRNA synthetase (ARS) involved in cancer development and proliferation [10]. ARSs have expression profiles similar to those of the first and second neighbor cancer-associated genes (CAGs) in 10 types of cancer, including PDAC, which are clearly distinguishable from the patterns of non-CAGs. Aberrant expression or post-translational modifications of ARSs are pathologically associated with cancers. The aminoacylation activity of MARS1, which is required for translation initiation, is increased in colon cancer [11]. Also, stable overexpression of the MARS1 substrate tRNA_i_^Met^ can cause oncogenic transformation [12]. Coincidently, the 3′-untranslated region (UTR) of MARS1 contains a 56-base-pair sequence complementary to the 3′-UTR of the C/EBP homologous protein, which is linked to the onset of certain tumors [13].

Overexpression of MARS1 has been reported in several cancer types, including malignant fibrous histiocytomas, sarcomas, malignant gliomas, and glioblastomas [14,15]. Elevated MARS1 expression is associated with a poor prognosis in non-small cell lung cancer (NSCLC) and breast cancer [16,17]. Suppression of MARS1 expression has been shown to reduce the cell transformation and tumorigenicity of p16INK4a-negative breast cancer cells [18]. MARS1 indirectly modulates tumor formation by interacting with AIMP3; thus, any mutation in MARS1 that interferes with its interaction with AIMP3 could modulate the tumor-suppressive activity of AIMP3 in the nucleus [19]. This implicates MARS1 in cancer development and progression; however, no study has evaluated the association between MARS1 and the prognosis of PDAC. We evaluated MARS1 expression and its ability to predict the prognosis of PDAC by performing an immunohistochemical (IHC) analysis of surgical specimens.

## 2. Materials and Methods

### 2.1. Study Design and Samples

In this retrospective study, tissues were obtained from patients who underwent surgical treatment for PDAC. The enrolled patients were eligible for surgery at the time of diagnosis, so neo-adjuvant chemotherapy was not performed, and adjuvant chemotherapy was performed depending on the patients’ condition after surgery. IHC analyses were conducted on tissue lysates and paraffin-embedded tissue blocks prepared using the surgical specimens. The MARS1 expression levels were evaluated in paired lysates from adjacent normal-appearing pancreatic and cancer-enriched tissues. The Ethics Committee of Gangnam Severance Hospital approved the study protocol. Written informed consent was obtained from the enrolled patients (IRB No. 3-2021-0444).

### 2.2. IHC Staining

Tumor tissues were fixed in 10% buffered formalin and embedded in paraffin. A 4 µm thick section from each paraffin block was subjected to IHC staining. The IHC analysis was performed using a primary antibody against human MARS1 (1:300; 0.2 mg/mL; Bicbio Inc., Suwon, South Korea) and an automated IHC stainer (BenchMark XT; Ventana Medical Systems, Tucson, AZ, USA).

### 2.3. Interpretation of IHC Staining

IHC staining was evaluated by two of the authors independently. MARS1 is strongly expressed in acinar cells (internal control cells) in normal pancreatic tissues, and weakly expressed in the benign pancreatic duct. This is because MARS1 expression is high in acinar tissue, which is present in both exocrine and endocrine organs, as a result of their high level of protein production. MARS1 expression is low in mucin-producing epithelia such as those in the pancreatic duct, which have low protein production. We used the MARS1 expression levels in acinar cells and the benign pancreatic duct as references. High MARS1 expression was defined as equal to, or greater than, that in normal acinar cells (Figure 1).

Low MARS1 expression was defined as weaker than that in normal acinar cells and stronger than that in benign pancreatic duct epithelium. To evaluate MARS1 expression, pancreatic cancer tissue was compared to normal pancreatic tissues, such as acinar cells and pancreatic duct epithelium. Figure 2 and Figure 3 show tissues with low and high MARS1 expression, respectively, as visualized by IHC and hematoxylin and eosin [H&E] staining of the same tissues.

The IHC results were interpreted in a manner blinded to the clinical data. All the enrolled patients had available follow-up and clinicopathological information, including age, tumor grade, tumor size, histological type, and tumor, node, metastasis (TNM) stage.

### 2.4. Statistical Analysis

Clinicopathological characteristics are presented as means ± standard deviations (SD) for continuous variables, or numbers (percentages) for categorical variables. To test differences according to MARS1 expression, the independent two-sample t-test was used for continuous variables, and the chi-squared test (Fisher’s exact test) for categorical variables. CA19-9 and CEA values are expressed as medians and interquartile ranges, and were analyzed using the Mann–Whitney U test. The Kaplan–Meier method, along with the log-rank test, was used to compare overall survival (OS) and disease-free survival (DFS) rates according to MARS1 expression. We used the z-test to evaluate the 2-year survival rate. The Cox proportional hazards regression model was used for univariate and multivariate analyses and to calculate adjusted hazard ratios (HR) and 95% confidence intervals (CI). Variables with a two-sided *p* < 0.05 in univariate analyses, as well as clinically important variables, were included in the multivariate analysis. We used SPSS software, version 20.0 (IBM Corp.; Armonk, NY, USA) for statistical analysis. The statistical tests were two sided, and *p* < 0.05 was considered indicative of statistical significance.

## 3. Results

### 3.1. Patients’ Characteristics

Surgical samples were collected from 137 patients with PDAC who underwent surgery between 2012 and 2022. Cancerous and adjacent normal pancreatic tissues were assessed for MARS1 expression (Table 1).

Of the patients, 55 and 82 had low and high MARS1 expression, respectively. The mean age of the low MARS1 expression group was 65.8 years, and that of the high MARS1 expression group was 65.9 years. The male–female ratio was similar in the two groups. There were no significant differences in the surgical methods (pylorus-preserving pancreaticoduodenectomy) between the two groups. The R0 resection rate was 70.1% in the low MARS1 expression group and 61% in the high MARS1 expression group (*p* = 0.112). The mean tumor size and TNM stage were not different between the two groups. The rate of moderate differentiation is lower in the high MARS1 expression group compared to the low MARS1 group, and there is a tendency for a higher rate of poor differentiation, although these trends were not statistically significant. No difference was found between the two groups in terms of lymph node metastasis, lymphovascular invasion, and perineural invasion in surgical tissue. Also, there were no differences in the mean CA 19-9 and CEA levels before surgery between the two groups. The rate of adjuvant chemotherapy after surgery was high in the low MARS1 expression group.

### 3.2. Overall and Disease-Free Survival

We analyzed the DFS and OS of the 137 patients with PDAC using the Kaplan–Meier method. DFS was defined as the time from surgery to first relapse, and OS as that from surgery to death. Patients with PDAC with high MARS1 expression had shorter DFS and OS than those with low MARS1 expression (Figure 4A,B).

Specifically, the median OS of the high MARS1 expression group was 15.2, versus 17.2 months in the low expression group (log-rank test *p* = 0.044). Although the median DFS was not significantly different between the two groups, the duration of DFS tended to be shorter in the high versus the low MARS1 expression group (8.9 versus 11.2 months, log-rank test *p* = 0.067). This suggests that the MARS1 expression level influences the OS and DFS of patients with PDAC. The 2-year survival rate is usually evaluated clinically in a meaningful way, so the 2-year survival rate was compared using a z-test (Appendix A). The low MARS1 expression group showed a significantly higher 2-year OS rate (*p* = 0.0345), but there was no significant difference between the two groups in the DFS rate (*p* = 0.0624) (Table 2).

### 3.3. Risk Factors for Overall and Recurrence-Free Survival

In univariate analyses, tumor size >3 cm (hazard ratio [HR] = 1.546, *p* = 0.015), TNM stage (HR = 2.052, *p* = 0.009), lymph node metastasis (HR = 2.310, *p* < 0.001), lymphovascular invasion (HR = 1.438, *p* = 0.041), perineural invasion (HR = 2.105, *p* = 0.001), and high MARS1 expression (HR = 5.663, *p* = 0.001, Table 3) were significantly associated with a poor OS rate of patients with PDAC. In the multivariate analysis, high MARS1 expression (HR = 2.761, *p* = 0.022) and lymph node metastasis (HR = 8.019, *p* = 0.048) were independent prognostic markers for poor OS.

A tumor size of >3 cm (HR = 1.770, *p* = 0.009) and high MARS1 expression (HR = 4.857, *p* = 0.005) were significantly associated with the DFS of patients with PDAC in univariate analyses (Table 2). High MARS expression (HR = 2.774, *p* = 0.023) and tumor size >3 cm (HR = 1.776, *p* = 0.010) were independent prognostic markers for poor DFS in the multivariate analysis. The multivariate analysis identified MARS1 expression as a significant and independent prognostic factor for OS and DFS. These results suggest that MARS1 expression has potential as a prognostic factor for patients with pancreatic cancer.

## 4. Discussion

We report that MARS1 expression was increased in pancreatic cancer tissue compared to normal pancreatic duct tissue. Patients with high MARS1 expression had shorter OS and DFS than those with low MARS1 expression.

MARS1 is a critical enzyme in translation initiation, transferring Met to the initiator tRNA [20]. MARS1 links the DNA damage response to global translation control after UV-mediated DNA damage; MARS1 is phosphorylated at Ser662 and dissociates from AIMP3. MARS1 is an ARS, which are housekeeping enzymes that catalyze amino acid ligation to their cognate transfer RNAs (tRNAs) with high precision, and thus are essential for protein biosynthesis [10]. ARS enzymes, which consume one molecule of ATP per reaction, activate amino acids to aminoacyl adenylates and deliver them to the acceptor ends of tRNAs. The overexpression of ARS may impact cancer survival and progression, and they have potential as anticancer therapeutics. The multifunctionality of ARSs and their localization to multiple regions suggests their potential as diagnostic biomarkers for cancer.

MARS1 promotes the carcinogenesis and progression of several human malignancies [14,15]. MARS1 reportedly has increased catalytic activity in human colon cancer [11], and stable overexpression of the MARS1 substrate tRNAiMet can cause oncogenic transformation [12]. Moreover, MARS1 stabilizes CDK4 by forming a complex with heat shock protein 90 in the cell division cycle, thus preventing the proteasome-dependent degradation of CDK4 [18]. Suppression of MARS1 expression reduces the cellular CDK4 level, resulting in cell cycle arrest at the G0/G1 phase [18]. MARS1 competes with p16^INK4a^, a tumor suppressor, for binding to the CDK4 N-terminal domain, further implicating it in p16^INK4a^-negative cancers [18]. Suppression of the expression of MARS1 reduced cell transformation and the tumorigenic ability of the p16^INK4a^-negative breast cancer cells [18], implicating MARS1 in the development and progression of cancer. However, its expression pattern and biological behavior in pancreatic carcinoma are unclear. Therefore, we investigated the clinical and prognostic utility of MARS1 in PDAC.

MARS1 is related to mTORC1 activity and is frequently overexpressed in NSCLC. MARS1 overexpression is associated with poor clinical outcomes, indicating its potential as a therapeutic target [16]. Autoantibodies against AIMP2-DX2 and AIMP2 are detectable in human blood, and an increased AIMP2-DX2/AIMP2 ratio is related to poor clinical outcomes in patients with lung cancer [21]. High MARS1 expression in human breast cancer tissues has been significantly associated with an unfavorable prognosis, suggesting MARS1 to have potential as a diagnostic and prognostic marker for breast cancer [17].

Elevated MARS1 expression in carcinoma can be used to diagnose cancer in indeterminate specimens [22]. The high sensitivity and accuracy of MARS1 immunofluorescence (IF) staining enables the detection of biliary malignancy in patients with an indeterminate biliary stricture. IF staining for MARS1 is more sensitive than conventional Pap staining (93.6 versus 73.2%, *p* < 0.001). Dual IF staining for MARS1/CD45 has shown good diagnostic performance, and could complement conventional cytologic tests for identifying LN metastasis in NSCLC [23]. The combination of MARS1 staining with conventional cytology has increased the diagnostic accuracy of computed tomography of the chest for lung nodules suspected of lung cancer. By complementing conventional cytology, dual IF staining for MARS1, AIMP2-DX2, and pan-CK improves the diagnostic yield of lung cancer [24].

In this study, the multivariate analysis showed that lymph node metastasis and high MARS1 expression were independent risk factors for OS. High MARS1 expression and tumor size >3 cm were independent risk factors for DFS. Elevated MARS1 expression is a poor prognostic factor for OS and DFS. The rate of adjuvant chemotherapy was 81.8% in the low MARS1 expression group and 58.5% in the high MARS1 expression group (*p* = 0.017). Adjuvant chemotherapy can affect PDAC recurrence and survival [2,5,6]. However, in this study, adjuvant chemotherapy was not linked to tumor recurrence or survival in either univariate or multivariate analyses.

MARS1 is an ARS and is involved in the development and proliferation of cancer [10]. MARS1 controls the methionylation of the initiator tRNA for translation initiation, regulates the initiation of protein synthesis, and enables cell cycle transitions [20]. Cancer cells have high translation rates and rapid cell cycle transitions [20]; therefore, pancreatic cancer with high MARS1 expression is likely to be aggressive. The result is a high recurrence rate after surgery and rapid progression, reducing the OS duration. Also, in cancer cells MARS1 might be present in a conformation, modification, or physical status different than that in normal cells, although this requires further in-depth investigation.

We speculate that high MARS1 expression is correlated with lymph node metastasis or tumor occurrence, because MARS1 is involved in cancer development and progression. Tumor stage is the primary prognostic factor for PDAC. Early-stage tumors are associated with longer survival than locally advanced or metastatic tumors because surgical resection is the only treatment for PDAC [3]. Only patients with operable PDAC were enrolled in this study; therefore, the TNM stage, including tumor size, did not affect the prognosis. However, node metastasis affected the OS. Lymph node positivity was the most significant risk factor in a postoperative prognostic PDAC model [25]. Resection status is an independent predictor of survival [4]. Lymph node positivity and an R1 resection margin are associated with postoperative recurrence, and OS is poor when PDAC recurs after surgery. Lymph node positivity and incomplete adjuvant chemotherapy are independent prognostic factors in resectable PDAC after neoadjuvant chemotherapy [26]. Further studies are needed to determine whether high MARS1 expression is linked to lymph node metastasis or tumor recurrence. MARS1 expression was relatively low in moderately differentiated PDAC, and relatively high in poorly differentiated PDAC. Although there is no statistical significance in this trend, further research is needed to explore the potential pathological relationship between PDAC differentiation type and MARS1 expression level, considering the role of MARS1 in cancer development.

CA 19-9 is a prognostic biomarker for PDAC; an elevated CA 19-9 level after surgery is predictive of PDAC recurrence [27]. Neoadjuvant chemotherapy is recommended for patients with a CA 19-9 level of >500 IU/L before surgery because of the risk of recurrence [28]. In this study, the CA 19-9 level at diagnosis was not predictive of the prognosis of PDAC. CA 19-9 possesses several limitations when interpreting serum levels in a clinical setting, and even provides a false negative in 5–20% of PDAC patients (those producing a specific sialylated antigen). Thus, CA 19-9 has limitations in effectively predicting PDAC prognoses.

The limitations of this study include, firstly, its retrospective design and the small population; a large-scale prospective study is needed to confirm the findings. Second, this study enrolled only patients with PDAC who underwent surgery. The proportion of patients with PDAC who undergo surgery at the time of diagnosis is < 20%; as such, whether MARS1 expression affects the prognosis of unresectable PDAC needs to be confirmed. (Interestingly, high MARS1 expression is linked to a poor prognosis in NSCLC and breast cancer.) Third, the standard for MARS1 expression is that found in normal tissue. However, normal tissue cannot be obtained through endoscopic ultrasound (EUS)-guided fine needle aspiration (FNA) or biopsy (FNB). EUS-FNA or EUS-FNB enables the effective collection of PDAC cells and tissues [29,30]. In order to make this marker clinically applicable to all patients with PDAC, a new cytology grading system for EUS-FNA or EUS-FNB, developed via further research, is needed to facilitate its application in unresectable PDAC. Fourth, we did not evaluate the mechanism underlying the link between high MARS1 expression and a poor prognosis of PDAC.

## 5. Conclusions

The expression level of MARS1 was significantly higher in PDAC tissues than in normal pancreatic duct tissues. Patients with PDAC with high MARS1 expression had shorter OS and DFS than those with low MARS1 expression. Additionally, lymph node metastasis, perineural invasion, and high MARS1 expression were independent risk factors for OS duration. Poor differentiation, R1 resection margin, and high MARS1 expression were independent risk factors for DFS duration. Our results implicate MARS1 in pancreatic carcinogenesis and suggest its potential as a novel prognostic marker for PDAC. Further prospective studies are needed to validate our findings.

## Figures and Tables

**Figure 1 cancers-15-05413-f001:**
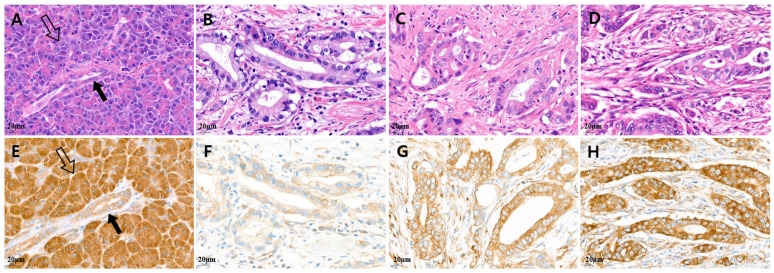
Representative microscopic features of MARS1 IHC expression in pancreatic adenocarcinoma. (**A**,**E**) Normal acinar cells (hollow arrow) showed strong MARS1 expression, and normal pancreatic duct epithelium (black arrow; internal control) showed moderate to weak MARS1 expression. ((**B**,**F**), (**C**,**G**)) Low MARS1 expression was defined as weaker than that in normal acinar cells, whereas (**D**,**H**) high MARS1 expression was defined as equal to, or stronger than, that in normal acinar cells ((**A**–**D**), H&E staining; (**E**–**H**), MARS1, ×400).

**Figure 2 cancers-15-05413-f002:**
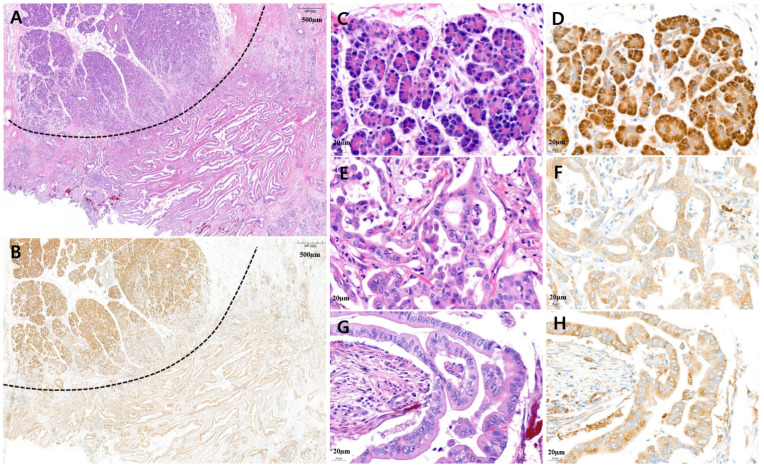
Representative microscopic features of low MARS1 immunohistochemistry (IHC) expression in pancreatic ductal adenocarcinoma (PDAC). (**A**) A representative surgical specimen showed normal pancreatic tissue (above dotted line) and PDAC tissue (below dotted line) (H&E staining, ×20). (**B**) MARS1 IHC staining represented low expression of MARS1 in the same tissue (MARS1, ×20). (**C**,**D**) The normal acinar cells showed strong MARS1 expression as an internal control ((**C**), H&E; (**D**), MARS1, ×400). (**E**–**H**) Pancreatic ductal adenocarcinoma—moderately differentiated—showed low MARS1 expression, which is weaker than MARS1 expression in normal acinar cells ((**E**,**G**), H&E staining; (**F**,**H**), MARS1, ×400).

**Figure 3 cancers-15-05413-f003:**
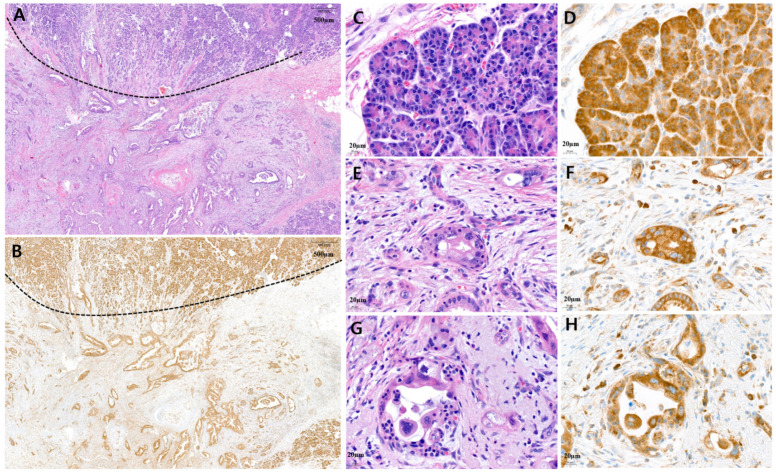
Representative microscopic features of high MARS1 immunohistochemistry (IHC) expression in pancreatic ductal adenocarcinoma (PDAC). (**A**) A representative surgical specimen showing normal pancreatic tissue (above dotted line) and PDAC tissue (below dotted line) (H&E staining, ×20). (**B**) MARS1 IHC staining represented high expression of MARS1 in same tissue (MARS1, ×20). (**C**,**D**) The normal acinar cells showed strong MARS1 expression as an internal control ((**C**), H&E; (**D**), MARS1, ×400). (**E**–**H**) Pancreatic ductal adenocarcinoma—moderately to poorly differentiated—showed high MARS1 expression, which is equal to or stronger than MARS1 expression in normal acinar cells ((**E**,**G**), H&E staining; (**F**,**H**), MARS1, ×400).

**Figure 4 cancers-15-05413-f004:**
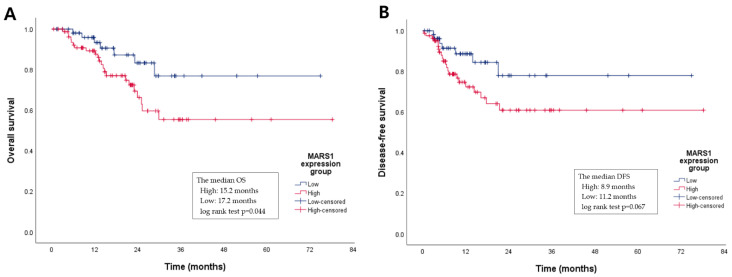
A Kaplan–Meier graph of the overall survival (OS) and disease-free survival (DFS) periods according to MARS1 expression. (**A**) The median OS period of the high MARS1 expression group is shorter than that of the low MARS1 expression group (15.2 vs. 17.2 months, log-rank test *p* = 0.044). (**B**) Although the median DFS between the two groups is not statistically different, the DFS trend in the high MARS1 expression group is also shorter than in the low group (8.9 vs. 11.2 months, log-rank test *p* = 0.067).

**Table 1 cancers-15-05413-t001:** Basic characteristics of patients with PDAC and MARS1 expression.

Variables	Low MARS1 Expression Group (n = 55)	High MARS1 Expression Group (n = 82)	*p*-Value
**Age**, y (mean ± SD)	65.8 ± 10.0	65.9 ± 10.3	0.543
**Sex**, n (M:F)	24:31	44:38	0.250
**Operation**, n (%)			0.311
PPPD	31 (56.4)	54 (66.9)	
Distal pancreatectomy	21 (38.2)	22 (26.9)	
Total pancreatectomy	3 (5.5)	6 (7.3)	
**Resection margin**, n (%)			
R0 resection	39 (70.1)	50 (61.0)	0.112
R1 resection	16 (29.9)	32 (39.0)	0.191
**Tumor size**, cm (mean ± SD)	3.2 ± 1.2	3.1 ± 1.3	0.552
**TNM stage**, n (%)			0.757
IA	2 (3.6)	3 (3.7)	
IB	5 (9.1)	7 (8.5)	
IIA	7 (12.7)	9 (11.0)	
IIB	22 (40.1)	42 (51.2)	
III	19 (34.5)	21 (25.6)	
**Differentiation**, n (%)			0.078
Well diff.	3 (5.5)	5 (6.1)	
Moderate diff.	50 (90.9)	63 (76.8)	
Poor diff.	2 (3.6)	14 (17.1)	
**Lymph node metastasis**, n (%)	41 (74.5)	62 (75.6)	0.889
**Lymphovascular invasion**, n (%)	29 (52.7)	40 (48.9)	0.653
**Perineural invasion**, n (%)	45 (81.8)	62 (75.6)	0.393
**CA 19-9** at adm, IU/L (mean ± SD)	639.4 ± 2161.8	368.5 ± 855.2	0.112
**CEA** at adm, IU/L (mean ± SD)	7.3 ± 19.4	4.3 ± 5.6	0.262
**Adjuvant chemotherapy**, n (%)	45 (81.8)	48 (58.5)	0.017

PDAC, pancreatic ductal adenocarcinoma; MARS1, methionyl-tRNA synthetase 1; SD, standard deviation; M, male; F, female; PPPD, pylorus-preserving pancreaticoduodenectomy; CA19-9, carbohydrate antigen 19-9; CEA, carcinoembryonic antigen.

**Table 2 cancers-15-05413-t002:** Overall survival and disease-free survival at 2 years according to MARS1 expression.

	Low MARS1 Expression Group	High MARS1 Expression Group	*p*-Value
	2 yr Survival Rate (se)	2 yr Survival Rate (95% CI)	
Overall survival	0.833 (0.067)	0.662 (0.068)	0.0345
Disease-free survival	0.779 (0.085)	0.608 (0.072)	0.0624

MARS1, methionyl-tRNA synthetase 1; se, standard error; CI, confidence intervals.

**Table 3 cancers-15-05413-t003:** Risk factors for overall survival and disease-free survival.

Factors	Overall Survival				Disease-Free Survival			
	Univariate Analysis		Multivariate Analysis		Univariate Analysis		Multivariate Analysis	
	HR (95% CI)	*p*-Value	HR (95% CI)	*p*-Value	HR (95% CI)	*p*-Value	HR (95% CI)	*p*-Value
**Gender** (male vs. female)	0.696 (0.492–0.985)	0.565			0.933 (0.606–1.437)	0.754		
**Age (y)** (≤50 vs. >50)	0.789 (0.435–1.433)	0.789			0.774 (0.361–2.374)	0.872		
**Tumor size (cm)** (≤3 vs. >3 cm))	1.546 (1.088–2.196)	0.015	2.042 (0.945–4.410)	0.069	1.770 (1.150–2.722)	0.009	1.776 (1.147–2.717)	0.010
**Differentiation** (WD, MD vs. PD)	0.924 (0.328–2.601)	0.251			1.422 (0.771–2.622)	0.051		
**TNM stage** (I vs. more II)	2.052 (1.194–3.524)	0.009	2.328 (0.993–5.457)	0.052	1.393 (0.558–3.427)	0.484		
**Lymph node metastasis** (positive/negative)	2.310 (1.519–3.512)	<.001	8.019 (1.022–62.951)	0.048	1.220 (0.715–2.084)	0.466		
**Lymph vascular invasion** (positive/negative)	1.438 (1.015–2.036)	0.041	1.870 (0.821–4.258)	0.136	1.534 (0.992–2.374)	0.055		
**Perineural invasion** (positive/negative)	2.105 (1.354–3.271)	0.001	1.605 (0.639–4.031)	0.314	1.440 (0.792–2.618)	0.231		
**R0 resection margin** (positive/negative)	1.221 (0.844–1.767)	0.288			1.587 (0.986–2.555)	0.057		
**Adjuvant chemotherapy** (positive vs. negative)	1.132 (0.802–1.596)	0.481			1.346 (0.922–1.966)	0.124		
**MARS1 expression** (high vs. low)	5.663 (2.016–15.906)	0.001	2.761 (1.159–6.576)	0.022	4.857 (1.597–14.773)	0.005	2.774 (1.554–4.950)	0.023

HR, hazard ratio; WD, well differentiated; MD, moderately differentiated; PD, poorly differentiated; MARS1, methionyl-tRNA synthetase 1.

## Data Availability

The data used in this study are available in this article.

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
