# Peer review of "Prediction of Prognosis in Pancreatic Cancer According to Methionyl-tRNA Synthetase 1 Expression as Determined by Immunohistochemical Staining"

_cancers, 2023, doi:10.3390/cancers15225413_

Round 1

Reviewer 1 Report (Previous Reviewer 2)

Comments and Suggestions for Authors

Dear authors,

Thanks so much for provided the amended version of your manuscript and for answering my queries. However, there is just one thing that has not been  properly performed: in the multivariate analyses according to OS and DFS, only statistically significant variables found in the univariate analysis must be included in the multivariate analysis to not understimate those clinico-pathologic variables highy significant. 

Please provide this analysis with only statisticaly significant variables from univariate analysis.

Comments on the Quality of English Language

It´s OK

Author Response

â–£ For Reviewer 1

We appreciate your thoughtful comments regarding our manuscript.

â–£ Evaluations

Comments by the reviewers:

Reviewer 1

Thanks so much for provided the amended version of your manuscript and for answering my queries. However, there is just one thing that has not been properly performed: in the multivariate analyses according to OS and DFS, only statistically significant variables found in the univariate analysis must be included in the multivariate analysis to not understimate those clinico-pathologic variables highy significant.

Please provide this analysis with only statisticaly significant variables from univariate analysis.

Answer) The multivariate model included variables significant (p < 0.05) in univariate models, as reviewer’s recommendation. The HR (95% CI) and p-value are reported for all variables included in the multivariate model. We have revised the Table 3 as follows:

And we also revised the Result section as follows;

3.3 Risk factors for overall and recurrence-free survival

In univariate analyses, tumor size > 3 cm (hazard ratio [HR] = 1.546, p = 0.015), TNM stage (HR = 2.052, p = 0.009), lymph node metastasis (HR = 2.310, p < 0.001), lymphovascular invasion (HR = 1.438, p = 0.041), perineural invasion (HR = 2.105, p = 0.001), and high MARS1 expression (HR = 5.663, p = 0.001, Table 2) were significantly associated with a poor OS of patients with PDAC. In the multivariate analysis, high MARS1 expression (HR = 2.761, p = 0.022) and lymph node metastasis (HR = 8.019, p = 0.048) were independent prognostic markers for a poor OS.

Tumor size > 3 cm (HR = 1.770, p = 0.009) and high MARS1 expression (HR = 4.857, p = 0.005) were significantly associated with the DFS of patients with PDAC in univariate analyses (Table 2). High MARS expression (HR = 2.774, p = 0.023) and tumor size > 3 cm (HR = 1.776, p = 0.010) were independent prognostic markers for a poor DFS in the multivariate analysis. The multivariate analysis identified MARS1 expression as a significant and independent prognostic factor for OS and DFS. These results suggest that MARS1 expression has potential as a prognostic factor for patients with pancreatic cancer.

And we also revised the Discussion section as follows;

In this study, the multivariate analysis showed that lymph node metastasis and high MARS1 expression were independent risk factors for OS. High MARS1 expression and tumor size > 3 cm were independent risk factors for DFS. Elevated MARS1 expression is a poor prognostic factor for OS and DFS. The rate of adjuvant chemotherapy was 81.8% in the low MARS1 expression group and 58.5% in the high MARS1 expression group (p = 0.017). Adjuvant chemotherapy can affect PDAC recurrence and survival [2,5,6]. However, in this study, adjuvant chemotherapy was not linked to tumor recurrence or survival in univariate and multivariate analyses.

Reviewer 2 Report (Previous Reviewer 3)

Comments and Suggestions for Authors

Hi dear Editor and Authors,
I noticed that this publish candidate is a re-submitssion research article, and after reading the previous review report, I can judge that most of my comments were adequately addressed and article quality was significant improved.

However, some of the language and gramma errors remained as the figure legend of Figure 1 still suffers of wrong tense. I would suggest a thorough language check for this paper.

Additional, an interesting observation is the un-balance of the Differentiation sub-section in Table 1 (90.9 vs 76.8 % in moderate and 3.6 vs 17.1 % in poor). While I understand this may be pathological related to the High MARS1 expression, and it may explain the significant difference in the overall survive rate, It will be beneficial to readers, as well as authors, to comment this point in the Result and Discussion sections, if not look deeper for the pathological link of  Differentiation to MARS1 expression level.

Otherwise this paper can be judged as equivalent quality for publishing. So I would recommend of a minor reversion before acceptance.

Best.

Comments on the Quality of English Language

Some of the language and gramma errors remained as the figure legend of Figure 1 still suffers of wrong tense. I would suggest a thorough language check for this paper.

Author Response

â–£ For Reviewer 2

We appreciate your thoughtful comments regarding our manuscript.

â–£ Evaluations

Comments by the reviewers:

Reviewer 2

I noticed that this publish candidate is a re-submitssion research article, and after reading the previous review report, I can judge that most of my comments were adequately addressed and article quality was significant improved.

  1. However, some of the language and gramma errors remained as the figure legend of Figure 1 still suffers of wrong tense. I would suggest a thorough language check for this paper.

Answer) The figure legend of Figure 1 has been revised and the English has been corrected according to your comment as followings;

Figure 1. Representative IHC features of PDAC with MARS1 expression. (A and E) Normal acinar cells (hollow arrow) showed strong MARS1 expression, and normal pancreatic duct epithelium (black arrow; internal control) showed moderate to weak MARS1 expression. (B and F, C and G) Low MARS1 expression was defined as weaker than that in normal acinar cells, whereas (D and H) high MARS1 expression was defined as equal to or stronger than that in normal acinar cells. (A-D, H&E staining; E-H, MARS1, ´400).’

  1. Additional, an interesting observation is the un-balance of the Differentiation sub-section in Table 1 (90.9 vs 76.8 % in moderate and 3.6 vs 17.1 % in poor). While I understand this may be pathological related to the High MARS1 expression, and it may explain the significant difference in the overall survive rate, It will be beneficial to readers, as well as authors, to comment this point in the Result and Discussion sections, if not look deeper for the pathological link of Differentiation to MARS1 expression level.

Answer) As the reviewer pointed out, the high MARS1 expression group exhibits a lower rate of moderate differentiation and a tendency towards higher rates of poor differentiation compared to the low MARS1 group. While this suggests that MARS1 expression is relatively low in PDAC with moderate differentiation and relatively high in PDAC with poor differentiation, the lack of statistical significance in this trend makes it challenging to establish a direct relationship between differentiation type and MARS1 expression level. Additionally, in multivariate analysis, differentiation did not emerge as a significant factor for overall survival

As the reviewer pointed out, the pathological link of differentiation to MARS1 expression level may be meaningful. MARS1 is an aminoacyl-tRNA synthetase (ARS), which catalyzes the coupling of amino acids to cognate transfer RNAs (tRNAs) and participates in cancer development and progression.1,2 MARS1 controls the methionylation of the initiator tRNA for translation initiation, regulates the initiation of protein synthesis, and enables cell cycle transitions.3 Cancer cells have high translation rates and rapid cell cycle transitions,3 and MARS1 is critical in translation regulation. MARS1 is strongly expressed in a variety of cancers, which is associated with a poor prognosis.4-9 Therefore, pancreatic cancer with high MARS1 expression is likely to be aggressive. The result is a high recurrence rate after surgery and rapid progression, reducing the OS duration.

Therefore, we have added this trend to the result section and discussion section as follows.

In Result section

 The rate of moderate differentiation is lower in the high MARS1 expression group compared to the low MARS1 group, and there is a tendency for a higher rate of poor differentiation, although these trends were not statistically significant.

In Discussion Section

MARS1 expression was relatively low in moderately differentiated PDAC and relatively high in poorly differentiated PDAC. Although there is no statistical difference in this trend, further research is needed to explore the potential pathological relationship between PDAC differentiation type and MARS1 expression level considering the role of MARS1 in cancer development

Reference

  1. Kim S, You S, Hwang D. Aminoacyl-tRNA synthetases and tumorigenesis: more than

housekeeping. Nat Rev Cancer 2011;11:708-18.

  1. Kwon NH, Fox PL, Kim S. Aminoacyl-tRNA synthetases as therapeutic targets. Nat Rev

Drug Discov 2019;18:629-50.

  1. Kwon NH, Kang T, Lee JY, et al. Dual role of methionyl-tRNA synthetase in the regulation of translation and tumor suppressor activity of aminoacyl-tRNA synthetase-interacting multifunctional protein-3. Proc Natl Acad Sci U S A 2011;108:19635-40.
  2. Forus A, Florenes VA, Maelandsmo GM, et al. The protooncogene CHOP/GADD153, involved in growth arrest and DNA damage response, is amplified in a subset of human sarcomas. Cancer Genet Cytogenet 1994;78:165-71.
  3. Nilbert M, Rydholm A, Mitelman F, et al. Characterization of the 12q13-15 amplicon in soft tissue tumors. Cancer Genet Cytogenet 1995;83:32-6.
  4. Palmer JL, Masui S, Pritchard S, et al. Cytogenetic and molecular genetic analysis of a pediatric pleomorphic sarcoma reveals similarities to adult malignant fibrous histiocytoma. Cancer Genet Cytogenet 1997;95:141-7.
  5. Reifenberger G, Ichimura K, Reifenberger J, et al. Refined mapping of 12q13-q15 amplicons in human malignant gliomas suggests CDK4/SAS and MDM2 as independent amplification targets. Cancer Res 1996;56:5141-5.
  6. Kim EY, Jung JY, Kim A, et al. Methionyl-tRNA synthetase overexpression is associated with poor clinical outcomes in non-small cell lung cancer. BMC Cancer 2017;17:467.
  7. Park SW, Kim SS, Yoo NJ, et al. Frameshift Mutation of MARS Gene Encoding an Aminoacyl-tRNA Synthetase in Gastric and Colorectal Carcinomas with Microsatellite Instability. Gut Liver 2010;4:430-1.

Otherwise this paper can be judged as equivalent quality for publishing. So I would recommend of a minor reversion before acceptance.

  1. Some of the language and gramma errors remained as the figure legend of Figure 1 still suffers of wrong tense. I would suggest a thorough language check for this paper.

Answer) The figure legend of Figure 1 has been revised and the English has been corrected according to your comment as followings;

Figure 1. Representative IHC features of PDAC with MARS1 expression. (A and E) Normal acinar cells (hollow arrow) showed strong MARS1 expression, and normal pancreatic duct epithelium (black arrow; internal control) showed moderate to weak MARS1 expression. (B and F, C and G) Low MARS1 expression was defined as weaker than that in normal acinar cells, whereas (D and H) high MARS1 expression was defined as equal to or stronger than that in normal acinar cells. (A-D, H&E staining; E-H, MARS1, ´400).’

I revised it again to match the submission format. The manuscript has been revised and the English has been corrected according to your comment. The English in this document has been checked by at least two professional editors, both native speakers of English. For a certificate, please see: http://www.textcheck.com/certificate/jubWYe

Reviewer 3 Report (Previous Reviewer 1)

Comments and Suggestions for Authors

I think that the AA reviewed the paper according to queries raised by both reviewers

Author Response

â–£ For Reviewer 3

I think that the AA reviewed the paper according to queries raised by both reviewers

Answer) We appreciate your thoughtful comments regarding our manuscript.

Reviewer 4 Report (New Reviewer)

Comments and Suggestions for Authors

Very interesting study. My only comment is that the series was a surgical series and this may create a bias as surgical patients are more "fit" and present less advanced disease. Could the authors comment on the potential implications on their findings also in other patients unfit for surgery?

The authors should also comment in the introduction or discussion that adequate samples for IHC could be obtained also with non surgical techniques, such as EUS-FNB (in this regard cite the recent MAs: PMID: 33481633 and PMID: 31031330)

Author Response

â–£ For Reviewer 4

We appreciate your thoughtful comments regarding our manuscript.

â–£ Evaluations

Very interesting study. My only comment is that the series was a surgical series and this may create a bias as surgical patients are more "fit" and present less advanced disease. Could the authors comment on the potential implications on their findings also in other patients unfit for surgery?

The authors should also comment in the introduction or discussion that adequate samples for IHC could be obtained also with non surgical techniques, such as EUS-FNB (in this regard cite the recent MAs: PMID: 33481633 and PMID: 31031330)

Answer) As pointed out by the reviewer, immunohistochemistry (IHC) staining is feasible in samples obtained by both EUS-FNA and EUS-FNB. In our study, MARS1 expression was originally based on normal tissue, including acinar cells and ductal epithelium. However, it is challenging to obtain normal tissue from EUS-guided FNA samples. To address this limitation, we propose a new scoring system for semi-quantifying MARS1 expression in EUS-FNA and EUS-FNB samples, with scores of 0 indicating no expression, 1 for low expression, 2 for moderate expression, and 3 for high expression. This grading system would be particularly useful for studies involving EUS-guided FNA samples. While a cytology grading system is already in use for other carcinomas, we plan to assess MARS1 expression in samples obtained from EUS-guided FNA or FNB in patients with unresectable PDAC. We acknowledge this as a limitation in our study

Third, the standard for MARS1 expression was that in normal tissue. However, normal tissue cannot be obtained through endoscopic ultrasound (EUS)-guided fine needle aspiration (FNA) or biopsy (FNB). To make this marker clinically applicable to all patients with PDAC, a new cytology grading system for EUS-FNA or EUS-FNB via further research is needed to facilitate its application in unresectable PDAC.

Round 2

Reviewer 1 Report (Previous Reviewer 2)

Comments and Suggestions for Authors The authors have provided all my queries that I considered crucial to deserve publication.   These include the amendent in the COX multivariate analysis that was very confusing in the beginning.      Therefore, the manuscript could be accepted accordingly.

Author Response

â–£ For Reviewer 1

We appreciate your thoughtful comments regarding our manuscript.

â–£ Evaluations

Comments by the reviewers:

The authors have provided all my queries that I considered crucial to deserve publication.   These include the amendent in the COX multivariate analysis that was very confusing in the beginning. Therefore, the manuscript could be accepted accordingly.

Answer) We appreciate your correct and detailed review of the statistics. Through your careful review, a more accurate analysis was possible and the meaning of the paper became clearer.

Reviewer 4 Report (New Reviewer)

Comments and Suggestions for Authors

The authors improved their manuscript but still the bibliography is not completely up-to-date. I think the paper would be improved with some citations of EUS-FNB papers, as suggested previously.

Author Response

â–£ For Reviewer 4

We appreciate your thoughtful comments regarding our manuscript.

â–£ Evaluations

Comments by the reviewers:

The authors improved their manuscript but still the bibliography is not completely up-to-date. I think the paper would be improved with some citations of EUS-FNB papers, as suggested previously.

Answer) We cited the EUS-FNB papers you previously presented as follows. We appreciate your recommendations to appropriate papers for up-to-date on this paper.

‘Third, the standard for MARS1 expression was that in normal tissue. However, normal tissue cannot be obtained through endoscopic ultrasound (EUS)-guided fine needle aspiration (FNA) or biopsy (FNB). EUS-FNA or EUS-FNB enables the effective collection of PDAC cells and tissues. [33,34] To make this marker clinically applicable to all patients with PDAC, a new cytology grading system for EUS-FNA or EUS-FNB via further research is needed to facilitate its application in unresectable PDAC.’

Reference

  1. Facciorusso, A.; Mohan, B.P.; Crino, S.F.; Ofosu, A.; Ramai, D.; Lisotti, A.; Chandan, S.; Fusaroli, P. Contrast-enhanced harmonic endoscopic ultrasound-guided fine-needle aspiration versus standard fine-needle aspiration in pancreatic masses: A meta-analysis. Expert review of gastroenterology & hepatology 2021, 15, 821-828.
  2. Facciorusso, A.; Bajwa, H.S.; Menon, K.; Buccino, V.R.; Muscatiello, N. Comparison between 22g aspiration and 22g biopsy needles for eus-guided sampling of pancreatic lesions: A meta-analysis. Endoscopic ultrasound 2020, 9, 167-+.

Round 3

Reviewer 4 Report (New Reviewer)

Comments and Suggestions for Authors

The revised manuscript is OK now. Thank you!

This manuscript is a resubmission of an earlier submission. The following is a list of the peer review reports and author responses from that submission.

Round 1

Reviewer 1 Report

Comments and Suggestions for Authors

As pancreatic cancer incidence keeps rising, a greater interest in finding prognostic markers and therapeutic targets also increases. I found that the present paper focus on a novel biomarkers, which was found of interest in other cancers and the results are very interesting. However I would recommmend clarification of 2 points: 1) Did any of there patients performed neo-adjuvant chemotherapy (NAC) ? This must be clarified as recent trials strongly recommend NAC in most if not all patients - As MARS1 expression was evaluated in surgical specimens it is important to clarify this point 2) Following the issue raised previously, will MARS1expression be possible in preoperative biopsies as most of these will not have tumour, acinar and ductal normal tissue. This might be a limitation to the use of this marker in clinical practice  

Author Response

â–£ For Reviewer 1

We appreciate your thoughtful comments regarding our manuscript.

â–£ Evaluations

Comments by the reviewers:

Reviewer 1

As pancreatic cancer incidence keeps rising, a greater interest in finding prognostic markers and therapeutic targets also increases. I found that the present paper focus on a novel biomarkers, which was found of interest in other cancers and the results are very interesting. However I would recommmend clarification of 2 points:

  1. Did any of there patients performed neo-adjuvant chemotherapy (NAC)? This must be clarified as recent trials strongly recommend NAC in most if not all patients - As MARS1 expression was evaluated in surgical specimens it is important to clarify this point

Answer) All patients enrolled in this study were eligible for surgery at the time of diagnosis, so neo-adjuvant chemotherapy was not performed. Some of the patients underwent adjuvant chemotherapy, but no patient received neoadjuvant chemotherapy. We have added the following text to the Methods section:

‘The enrolled patients were eligible for surgery at the time of diagnosis, so neo-adjuvant chemotherapy was not performed, and adjuvant chemotherapy was performed depending on the patients’ condition after surgery.’

  1. Following the issue raised previously, will MARS1 expression be possible in preoperative biopsies as most of these will not have tumour, acinar and ductal normal tissue. This might be a limitation to the use of this marker in clinical practice  

Answer) Indeed, this marker has limitations in endoscopic ultrasound-guided fine needle aspiration (EUS-guided FNA). In this study, MARS1 expression was based on that in normal tissue including acinar cells and ductal epithelium. This is because it is difficult to obtain normal tissue from samples obtained by EUS-guided FNA. We plan to evaluate MARS1 expression in samples obtained by EUS-guided FNA. We have added the above as a limitation as follows:

‘Third, the standard for MARS1 expression was that in normal tissue. However, normal tissue cannot be obtained by endoscopic ultrasound (EUS)-guided fine needle aspiration (FNA). For this marker to be used clinically in all patients with PDAC, further research is needed to enable its application in conjunction with EUS-guided FNA.’

Reviewer 2 Report

Comments and Suggestions for Authors

The article entitled “Prognosis prediction in pancreatic cancer according to methionyl-tRNA synthetase 1 levels determined by immunohistchemical staining” by Sung Ill Jang et al., is overall well written; however, I strongly recommend an English review for clarity, grammar and usage.

Please find below my points to improve the quality of the manuscript.

-The article is presented without a format revision, please amend this.

-The manuscript is well introduced; however, I miss more data about the clinical management of PDAC patients.

-Please define the cut-off point to stratify patients between high and low expression. Did you use a Hscore?

-All images lack scales.

-Statistics are no properly described. Please revise this section with a statistician.

-Please include the type of neoadjuvant treatment used for borderline resectable patients included in the study.

-What is the significance in the manuscript of the statistically significant variable “Adjuvant treatment” in table 1.

-Is rather astonishing that median PFS or OS was not reached. Please include what is the 5 years overall survival of PDAC patients in Korea. Maybe other tests suits better than log-rank. 

-Why multivariate analyses of OS is only performed with 0.001 significant variables? Vascular invasion, TNM, tumour size should also be included. However, multi-variate analysis of PFS includes differentiation that was not significant in the univariate.

-A potential mechanisms should be included.

Comments on the Quality of English Language

 I strongly recommend an English review for clarity, grammar and usage.

Author Response

â–£ For Reviewer 2

We appreciate your thoughtful comments regarding our manuscript.

â–£ Evaluations

Comments by the reviewers:

Reviewer 2

The article entitled “Prognosis prediction in pancreatic cancer according to methionyl-tRNA synthetase 1 levels determined by immunohistchemical staining” by Sung Ill Jang et al., is overall well written; however, I strongly recommend an English review for clarity, grammar and usage.

Please find below my points to improve the quality of the manuscript.

  1. The article is presented without a format revision, please amend this.

Answer) I revised it again to match the submission format. The manuscript has been revised and the English has been corrected according to your comment. The English in this document has been checked by at least two professional editors, both native speakers of English. For a certificate, please see: http://www.textcheck.com/certificate/jubWYe

  1. The manuscript is well introduced; however, I miss more data about the clinical management of PDAC patients.

Answer) We have added this information to the Introduction section as follows:

‘Resectable PDAC is first treated surgically, followed by adjuvant chemotherapy [2-4]. For borderline resectable PDAC, neo-adjuvant chemotherapy is followed by surgery or continuing chemotherapy; palliative chemotherapy is performed for unresectable PDAC.’

  1. Please define the cut-off point to stratify patients between high and low expression. Did you use a Hscore?

Answer) The MARS1 expression levels were based on those in normal tissues, acinar cells, and pancreatic duct epithelium. Normal tissues such as acinar cells and pancreatic duct epithelium express, but do not overexpress, MARS1. However, because acinar tissue are glands, they have higher MARS1 expression than normal pancreatic-duct epithelium. PDAC originates from the pancreatic-duct epithelium and so has higher MARS1 expression than normal pancreatic-duct epithelium. The MARS1 standard for PDAC was determined based on the differences in MARS1 expression levels among tissues; the H-score was not used. We have revised the text as follows:

‘MARS1 is strongly expressed in acinar cells (internal control cells) in normal pancreatic tissues, and weakly expressed in the benign pancreatic duct. This is because MARS1 expression is high in acinar tissue, which are exocrine and endocrine organ, as a result of their high level of protein production. MARS1 expression is low in mu-cin-producing epithelia such as those in the pancreatic duct, which have low protein production. We used the MARS1 expression levels in acinar cells and the benign pan-creatic duct as references. High MARS1 expression was defined as equal to or stronger than that in normal acinar cells.’

  1. All images lack scales.

Answer) We have added scale bars to all the images (red circles).

Figure 1

Figure 2

Figure 3

  1. Statistics are no properly described. Please revise this section with a statistician.

Answer) The statistical method has been modified as follows:

Clinicopathological characteristics are presented as means ± standard deviations (SD) for continuous variables or numbers (percentages) for categorical variables. To test differences according to MARS1 expression, the independent two-sample t-test was used for continuous variables and the chi-squared test (Fisher’s exact test) for categorical variables. CA19-9 and CEA values are expressed as medians and inter-quartile ranges and were analyzed using the Mann–Whitney U-test. The Kaplan–Meier method with the log-rank test was used to compare overall survival (OS) and dis-ease-free survival (DFS) rates according to MARS1 expression. We used the z-test to evaluate the 2-year survival rate. The Cox proportional hazards regression model was used for univariate and multivariate analyses and to calculate adjusted hazard ratios (HR) and 95% confidence intervals (CI). Variables with a two-sided p < 0.05 in univariate analyses as well as clinically important variables were included in the multivariate analysis. We used SPSS software, version 20.0 (IBM Corp.; Armonk, NY, USA) for statistical analysis. The statistical tests were two sided, and p < 0.05 was considered indicative of statistical significance.

  1. Please include the type of neoadjuvant treatment used for borderline resectable patients included in the study.

Answer) All patients enrolled in this study were eligible for surgery at the time of diagnosis, so neo-adjuvant chemotherapy was not performed. Some of the patients underwent adjuvant chemotherapy, but no patient received neo-adjuvant chemotherapy. We have added the following text to the Methods section:

‘The enrolled patients were eligible for surgery at the time of diagnosis, so neoadjuvant chemotherapy was not performed, and adjuvant chemotherapy was performed de-pending on the patients’ condition after surgery.’

  1. What is the significance in the manuscript of the statistically significant variable “Adjuvant treatment” in table 1.

Answer) Some of the enrolled patients underwent adjuvant chemotherapy, but no patient received neo-adjuvant chemotherapy. We have revised the Methods section as follows:

‘The enrolled patients were eligible for surgery at the time of diagnosis, so neo-adjuvant chemotherapy was not performed, and adjuvant chemotherapy was performed depending on the patients’ condition after surgery.’

The rate of adjuvant chemotherapy was 81.8% in the low MARS1 expression group and 58.5% in the high MARS1 expression group (p = 0.017; Table 1). Adjuvant chemotherapy can affect tumor recurrence and survival duration in patients with PDAC, but in this study it did not affect tumor recurrence or survival duration in univariable and multivariable analyses. The Discussion section has been revised as follows:

‘The rate of adjuvant chemotherapy was 81.8% in the low MARS1 expression group and 58.5% in the high MARS1 expression group (p = 0.017). Adjuvant chemotherapy can affect PDAC recurrence and survival [2,5,6]. However, in this study, adjuvant chemotherapy was not linked to tumor recurrence or survival in univariate and multivariate analyses.’

  1. Is rather astonishing that median PFS or OS was not reached. Please include what is the 5 years overall survival of PDAC patients in Korea. Maybe other tests suits better than log-rank. 

Answer) According to data from the South Korea National Cancer Center, the 5-year OS of PDAC in South Korea is 15.2%. The OS of PDAC for which surgery is possible is 48%, and that of PDAC for which surgery is not possible is 20.4% (local advanced PDAC) and 2.4% (unresectable PDAC).

The log-rank test, which assigns the same weight at each time point and evaluates differences between groups, was important, so we maintained the results and compared survival rates in specific years. However, few patients were followed-up for 5 years, so we evaluated the 2-year survival rate by z-test. The low MARS1 expression group had a significantly longer OS than the high MARS1 expression group (p = 0.0345), but there was no significant difference in DFS between the two groups (p = 0.0624). The following text has been added to the Results section:

‘The 2-year survival rate is usually evaluated clinically in a meaningful way, so the 2-year survival rate was compared using z-test. The low MARS1 expression group showed a significantly higher 2-year OS rate (p = 0.0345), but there was no significant difference between the two groups in the DFS rate (p = 0.0624) (Table 2).’

Table 2. Overall survival and disease free survival at 2 years according to MARS1 expression.

Low MARS1

expression group

High MARS1

expression group

p-value

2yr survival rate (se)

2yr survival rate (95% CI)

Overall survival

0.833(0.067)

0.662(0.068)

0.0345

Disease free survival

0.779(0.085)

0.608(0.072)

0.0624

MARS1, methionyl-tRNA synthetase 1; se, standard error; CI, confidence interval

  1. Why multivariate analyses of OS is only performed with 0.001 significant variables? Vascular invasion, TNM, tumour size should also be included. However, multi-variate analysis of PFS includes differentiation that was not significant in the univariate.

Answer) The multivariate model included variables significant (p < 0.05) in univariate models as well as clinically important variables. The HR (95% CI) and p-value are reported for all variables included in the multivariate model.

Factors

Overall survival

Disease-free survival

Univariate analysis

Multivariate analysis

Univariate analysis

Multivariate analysis

HR (95% CI)

P-value

HR (95% CI)

P-value

HR (95% CI)

P-value

HR (95% CI)

P-value

Gender

(male vs. female)

0.696   (0.492–0.985)

0.565

1.779    (0.725-4.366)

0.209

0.933    (0.606–1.437)

0.754

1.855    (0.835-4.121)

0.129

Age (y)

(≤50 vs. >50)

0.789   (0.435–1.433)

0.789

0.344    (0.079-1.500)

0.156

0.774    (0.361–2.374)

0.872

1.152    (0.281-4.723)

0.844

Tumor size (cm)

(≤3 vs. >3 cm)

1.546   (1.088–2.196)

0.015

1.337    (0.519-3.446)

0.548

1.770   (1.150–2.722)

0.009

0.840    (0.369-1.911)

0.677

Differentiation

(WD, MD vs. PD)

0.924   (0.328–2.601)

0.251

0.962    (0.245-3.774)

0.956

1.422   (0.771–2.622)

0.051

1.972   (1.026–3.788)

0.042

TNM stage

(I vs. more II)

2.052   (1.194–3.524)

0.009

0.220    (0.014-3.455)

0.281

1.393   (0.558–3.427)

0.484

0.191    (0.036-1.004)

0.151

Lymph node metastasis (positive/negative)

2.310   (1.519–3.512)

<.001

2.047   (1.122–3.735)

0.020

1.220   (0.715–2.084)

0.466

1.548    (0.408-5.879)

0.521

Lymph vascular invasion (positive/negative)

1.438   (1.015–2.036)

0.041

0.512    (0.188-1.397)

0.191

1.534   (0.992–2.374)

0.055

0.725    (0.316-1.662)

0.448

Perineural invasion (positive/negative)

2.105   (1.354–3.271)

0.001

1.887   (1.129–3.154)

0.015

1.440   (0.792–2.618)

0.231

0.501    (0.184-1.367)

0.177

R0 resection margin (positive/negative)

1.221   (0.844–1.767)

0.288

0.581    (0.212-1.588)

0.289

1.587   (0.986–2.555)

0.057

1.750   (1.036–2.955)

0.036

Adjuvant chemotherapy (positive/negative)

1.132   (0.802–1.596)

0.481

0.952    (0.344-2.633)

0.924

1.346   (0.922–1.966)

0.124

0.576    (0.244-1.360)

0.576

MARS1 expression

(High vs. low)

5.663   (2.016–15.906)

0.001

4.532   (1.496–13.729)

0.008

4.857   (1.597–14.773)

0.005

4.130   (1.348–12.660)

0.013

  1. A potential mechanisms should be included.

Answer) MARS1 is an aminoacyl-tRNA synthetase (ARS), which catalyzes the coupling of amino acids to cognate transfer RNAs (tRNAs) and participates in cancer development and progression.1,2 MARS1 controls the methionylation of the initiator tRNA for translation initiation, regulates the initiation of protein synthesis, and enables cell cycle transitions.3 Cancer cells have high translation rates and rapid cell cycle transitions,3 and MARS1 is critical in translation regulation. MARS1 is strongly expressed in a variety of cancers, which is associated with a poor prognosis.4-9 Therefore, pancreatic cancer with high MARS1 expression is likely to be aggressive. The result is a high recurrence rate after surgery and rapid progression, reducing the OS duration. This has been added to the Discussion section.

‘MARS1 is an ARS and is involved in the development and proliferation of cancer [10,11]. MARS1 controls the methionylation of the initiator tRNA for translation initiation, regulates the initiation of protein synthesis, and enables cell cycle transitions [21]. Cancer cells have high translation rates and rapid cell cycle transitions [21]; therefore, pancreatic cancer with high MARS1 expression is likely to be aggressive. The result is a high recurrence rate after surgery and rapid progression, reducing the OS duration. Also, in cancer cells MARS1 might be present in a conformation, modification, or physical status different from that in normal cells, although this requires further in-depth investigation.’

Reference

  1. Kim S, You S, Hwang D. Aminoacyl-tRNA synthetases and tumorigenesis: more than

housekeeping. Nat Rev Cancer 2011;11:708-18.

  1. Kwon NH, Fox PL, Kim S. Aminoacyl-tRNA synthetases as therapeutic targets. Nat Rev

Drug Discov 2019;18:629-50.

  1. Kwon NH, Kang T, Lee JY, et al. Dual role of methionyl-tRNA synthetase in the regulation of translation and tumor suppressor activity of aminoacyl-tRNA synthetase-interacting multifunctional protein-3. Proc Natl Acad Sci U S A 2011;108:19635-40.
  2. Forus A, Florenes VA, Maelandsmo GM, et al. The protooncogene CHOP/GADD153, involved in growth arrest and DNA damage response, is amplified in a subset of human sarcomas. Cancer Genet Cytogenet 1994;78:165-71.
  3. Nilbert M, Rydholm A, Mitelman F, et al. Characterization of the 12q13-15 amplicon in soft tissue tumors. Cancer Genet Cytogenet 1995;83:32-6.
  4. Palmer JL, Masui S, Pritchard S, et al. Cytogenetic and molecular genetic analysis of a pediatric pleomorphic sarcoma reveals similarities to adult malignant fibrous histiocytoma. Cancer Genet Cytogenet 1997;95:141-7.
  5. Reifenberger G, Ichimura K, Reifenberger J, et al. Refined mapping of 12q13-q15 amplicons in human malignant gliomas suggests CDK4/SAS and MDM2 as independent amplification targets. Cancer Res 1996;56:5141-5.
  6. Kim EY, Jung JY, Kim A, et al. Methionyl-tRNA synthetase overexpression is associated with poor clinical outcomes in non-small cell lung cancer. BMC Cancer 2017;17:467.
  7. Park SW, Kim SS, Yoo NJ, et al. Frameshift Mutation of MARS Gene Encoding an Aminoacyl-tRNA Synthetase in Gastric and Colorectal Carcinomas with Microsatellite Instability. Gut Liver 2010;4:430-1.

Reviewer 3 Report

Comments and Suggestions for Authors

Hi dear Editor and Authors,

Thanks for inviting me to review the manuscript titled as "Prognosis prediction in pancreatic cancer according to methionyl-tRNA synthetase 1 levels determined by immunohistochemical staining". This research is properly designed, evidence is solid and conclusion is clear. The early diagnose of PDAC is the bottleneck of clinical practice and this research can contribute to this issue and benefit patience.

After reviewing the paper, I would like to suggest following comments to improve this manuscript.

  1. Please add more recent study on the MARS1 in PDAC in the Introduction (Line 66 to Line 77) and Discussion part. The author provided a background information regarding to expression pattern change of MARS1 in several cancer types. And a further illustration of an active role of MARS1 in PDAC will strongly support authors claim that MARS1 can be a biomarker of PDAC.
  2. Please adopt past tenses in the figure legends rather than primary tenses.
  3. For Table 1, CA 19-9 in Low MARS1 expression group is 639.4±2161.8 (mean±SD). The SD is too high, even compared to mean, and resulted in P value is 0.112. I would suggest a further data analyze to be done by either outlier removal or Z-Score Normalization, and illustrate in a supplemental table. https://www.graphpad.com/guides/prism/latest/curve-fitting/reg_oultlier.htm.
  4. For Table 1, CEA in Low MARS1 expression group is 7.3±19.4 (mean±SD), same as above. By doing so, a significant difference might be achieved in the Line 161 to 173.
  5. Is it possible to adopt a Scoring System to semi-quantify the expression of MARS1 in pancreatic ducts, saying 0 is no expression; 1 is low expression; 2 is moderate expression; and 3 is high expression? Adopting a scoring system can zoom in the difference between groups and enable a stratification analysis, taking the most advantage of the data.
  6. In Part 3.2 Risk factors, numbers of high MARS1 expression are change ((HR=5.663, p=0.001 vs HR=4.532, p=0.008) and confusing. I understand they are from two difference analyze tool, but it will be easier for readers to access by clarify them in the paragraph.

In summary, these valuable data can be revised to digging more information from them. I would suggest to accept this paper after minor modification.

Best

Comments on the Quality of English Language
  1. Please adopt past tenses in the figure legends rather than primary tenses.

Author Response

â–£ For Reviewer 3

We appreciate your thoughtful comments regarding our manuscript.

â–£ Evaluations

Comments by the reviewers:

Reviewer 3

Thanks for inviting me to review the manuscript titled as "Prognosis prediction in pancreatic cancer according to methionyl-tRNA synthetase 1 levels determined by immunohistochemical staining". This research is properly designed, evidence is solid and conclusion is clear. The early diagnose of PDAC is the bottleneck of clinical practice and this research can contribute to this issue and benefit patience.

After reviewing the paper, I would like to suggest following comments to improve this manuscript.

  1. Please add more recent study on the MARS1 in PDAC in the Introduction (Line 66 to Line 77) and Discussion part. The author provided a background information regarding to expression pattern change of MARS1 in several cancer types. And a further illustration of an active role of MARS1 in PDAC will strongly support authors claim that MARS1 can be a biomarker of PDAC.

Answer) This study is the first clinical study on MARS1 expression in PDAC. Association studies on MARS1 and tumorigenesis have been studied in previous studies as follows.

‘Aminoacyl-tRNA synthetases are essential enzymes that catalyze the coupling of amino acids to cognate transfer RNAs (tRNAs).1 Aminoacyl-tRNA synthetases are involved in a variety of cellular processes and play an important role in cancer development and progression.1 1,295 CAGs that are linked to the first ARS-neighbours (second neighbours) using the US National Cancer Institute (NCI) cancer gene index and 1,874 non-cancer associated genes (non-CAGs) is shown. Cancer-associated differential expression of each gene group was indicated by P values (left column) and fold changes (right column). The P values and the fold changes were sorted in ascending order. The P values are displayed in a red color gradient: dark red (P <0.01), red (0.01< P <0.05) and white (P >0.05). The fold changes are represented by a red–green gradient showing cancer-specific increased expression and decreased expression, respectively. The comparison of copy number variations of ARS and AIMP-encoding genes, first, second neighbour CAGs and non-CAGs is shown. Copy number variations in the nine cancer types were obtained from the CanGEM database. For each gene group, the relative frequency of copy number variations in each cancer type is presented (red circle at pancreatic cancer). HLT, haematopoietic and lymphoid tissue (figure 1).1

Figure 1 Expression profiles and copy number variations of ARS-encoding genes in comparison to CAGs and non-CAGs.

Moreover, a heat map shows the association of the individual ARSs and AIMPs with ten types of cancer. The cancer association score for each of the ARSs and AIMPs reflects the degree of deregulation in the corresponding cancers including pancreatic cancer (red circle) (Figure 2). 1

Figure 2. Hypothetical network model showing connections of ARSs with their CAG interactors.

Reference

  1. Kim S, You S, Hwang D. Aminoacyl-tRNA synthetases and tumorigenesis: more than housekeeping. Nat Rev Cancer 2011;11:708-18.

Based on this, as pointed out, we have added the background information for MARS1 as follows.

In Introduction section

MARS1 is an aminoacyl-tRNA synthetase (ARS) involved in cancer development and proliferation [10,11]. ARS have expression profiles similar to those of the first and sec-ond neighbor cancer-associated genes (CAGs) in 10 types of cancer, including PDAC, which are clearly distinguishable from the patterns of non-CAGs. Aberrant expression or post-translational modifications of ARS are pathologically associated with cancers.

In Discussion section

‘MARS1 is an ARS and is involved in the development and proliferation of cancer [10,11]. MARS1 controls the methionylation of the initiator tRNA for translation initiation, regulates the initiation of protein synthesis, and enables cell cycle transitions [21]. Cancer cells have high translation rates and rapid cell cycle transitions [21]; therefore, pancreatic cancer with high MARS1 expression is likely to be aggressive. The result is a high recurrence rate after surgery and rapid progression, reducing the OS duration. Also, in cancer cells MARS1 might be present in a conformation, modification, or physical status different from that in normal cells, although this requires further in-depth investigation.’

Reference

  1. Kim S, You S, Hwang D. Aminoacyl-tRNA synthetases and tumorigenesis: more than

housekeeping. Nat Rev Cancer 2011;11:708-18.

  1. Kwon NH, Fox PL, Kim S. Aminoacyl-tRNA synthetases as therapeutic targets. Nat Rev

Drug Discov 2019;18:629-50.

  1. Kwon NH, Kang T, Lee JY, et al. Dual role of methionyl-tRNA synthetase in the regulation of translation and tumor suppressor activity of aminoacyl-tRNA synthetase-interacting multifunctional protein-3. Proc Natl Acad Sci U S A 2011;108:19635-40.
  2. Forus A, Florenes VA, Maelandsmo GM, et al. The protooncogene CHOP/GADD153, involved in growth arrest and DNA damage response, is amplified in a subset of human sarcomas. Cancer Genet Cytogenet 1994;78:165-71.
  3. Nilbert M, Rydholm A, Mitelman F, et al. Characterization of the 12q13-15 amplicon in soft tissue tumors. Cancer Genet Cytogenet 1995;83:32-6.
  4. Palmer JL, Masui S, Pritchard S, et al. Cytogenetic and molecular genetic analysis of a pediatric pleomorphic sarcoma reveals similarities to adult malignant fibrous histiocytoma. Cancer Genet Cytogenet 1997;95:141-7.
  5. Reifenberger G, Ichimura K, Reifenberger J, et al. Refined mapping of 12q13-q15 amplicons in human malignant gliomas suggests CDK4/SAS and MDM2 as independent amplification targets. Cancer Res 1996;56:5141-5.
  6. Kim EY, Jung JY, Kim A, et al. Methionyl-tRNA synthetase overexpression is associated with poor clinical outcomes in non-small cell lung cancer. BMC Cancer 2017;17:467.
  7. Park SW, Kim SS, Yoo NJ, et al. Frameshift Mutation of MARS Gene Encoding an Aminoacyl-tRNA Synthetase in Gastric and Colorectal Carcinomas with Microsatellite Instability. Gut Liver 2010;4:430-1.

  1. Please adopt past tenses in the figure legends rather than primary tenses.

Answer) As you pointed out, the figure legends were modified to past tenses as shown below.

‘Figure 1. Representative microscopic features of MARS1 IHC expression in pancreatic adenocarcinoma. (A and E) Normal acinar cells (hollow arrow) showed strong MARS1 expression and normal pancreatic duct epithelium (black arrow) shows moderate to weak MARS1 expression as an internal control. (B and F, C and G) Low MARS1 expression was weaker than MARS1 expression in normal acinar cells, whereas (D and H) high MARS1 expression equal to or stronger than MARS1 expression in normal acinar cells. (A-D, H&E staining; E-H, MARS1, x400).’

‘Figure 2. Representative microscopic features of low MARS1 immunohistochemistry(IHC) expres-sion in pancreatic ductal adenocarcinoma (PDAC). (A) A representative surgical specimen showed normal pancreatic tissue (above dotted line) and PDAC tissue (below dotted line) (H&E staining, x20). (B) MARS1 IHC staining represented low expression of MARS1 in same tissue (MARS1, x20). (C and D) The normal acinar cells showed strong MARS1 expression as an internal control (C, H&E; D, MARS1, x400). (E, F, G and H) Pancreatic ductal adenocarcinoma, moderately differentiated showed low MARS1 expression which is weaker than MARS1 expression in normal acinar cells (E and G, H&E; F and H, MARS1, x400).’

‘Figure 3. Representative microscopic features of high MARS1 immunohistochemistry(IHC) ex-pression in pancreatic ductal adenocarcinoma (PDAC). (A) A representative surgical specimen showed normal pancreatic tissue (above dotted line) and PDAC tissue (below dotted line) (H&E staining, x20). (B) MARS1 IHC staining represented high expression of MARS1 in same tissue (MARS1, x20). (C and D) The normal acinar cells showed strong MARS1 expression as an internal control (C, H&E; D, MARS1, x400). (E, F, G and H) Pancreatic ductal adenocarcinoma, moder-ately to poorly differentiated showed high MARS1 expression which is equal to or stronger than MARS1 expression in normal acinar cells (E and G, H&E; F and H, MARS1, x400).’

‘Figure 4. A Kaplan–Meier graph of the overall and disease-free survival (DFS) according to MARS1 expression. (A) The median OS of the high MARS1 expression group was shorter than that of the low MARS1 expression group (15.2 vs. 17.2 months, log-rank test p=0.044). (B) Although the median DFS between the two groups was not statistically different, the DFS trend in the high MARS1 expression group was also shorter than in the low group (8.9 vs. 11.2 months, log-rank test p=0.067).’

  1. For Table 1, CA 19-9 in Low MARS1 expression group is 639.4±2161.8 (mean±SD). The SD is too high, even compared to mean, and resulted in P value is 0.112. I would suggest a further data analyze to be done by either outlier removal or Z-Score Normalization, and illustrate in a supplemental table. https://www.graphpad.com/guides/prism/latest/curve-fitting/reg_oultlier.htm.

Answer) A boxplot was generated to assess the CA19-9 distribution. To compensate for the very large standard deviation, the analysis was performed after removing outliers and changing the z-score; however, the standard deviation remained greater than the mean. Therefore, variables are expressed as medians and IQRs and were evaluated by non-parametric Mann–Whitney U-test (Figure 1, Table 1).

Figure 1. CA 19-9 and CEA level analyzed using Z-score

Table 1. Comparison of CA 19-9 and CEA level between two group

Variables

Non-parametric method

Low MARS1
expression group  (n=55)

High MARS1
expression group (n=82)

p-value

CA 19-9 at adm, IU/L (median(IQR))

97.80(25.30, 408.00)

58.80(14.50, 373.80)

0.437

CEA at adm, IU/L (median(IQR))

3.98(1.40, 6.80)

2.79(1.75, 4.60)

0.2771

This information has been added to the supplementary

Supplementary figure 1. CA 19-9 and CEA level analyzed using Z-score

Supplementary table 1 Comparison of CA 19-9 and CEA level between two group

Variables

Non-parametric method

Low MARS1
expression group  (n=55)

High MARS1
expression group (n=82)

p-value

CA 19-9 at adm, IU/L (median(IQR))

97.80(25.30, 408.00)

58.80(14.50, 373.80)

0.437

CEA at adm, IU/L (median(IQR))

3.98(1.40, 6.80)

2.79(1.75, 4.60)

0.2771

  1. For Table 1, CEA in Low MARS1 expression group is 7.3±19.4 (mean±SD), same as above. By doing so, a significant difference might be achieved in the Line 161 to 173.

Answer) Similar to review opinion at number 3, the standard deviation still tended to be large even after removing outliers or changing the Z-score, so the data was expressed in Median and IQR and tested with the Mann-whitney U test.

  1. Is it possible to adopt a Scoring System to semi-quantify the expression of MARS1 in pancreatic ducts, saying 0 is no expression; 1 is low expression; 2 is moderate expression; and 3 is high expression? Adopting a scoring system can zoom in the difference between groups and enable a stratification analysis, taking the most advantage of the data.

Answer) The MARS1 expression levels were based on those in normal tissues, acinar cells, and pancreatic duct epithelium. Normal tissues such as acinar cells and pancreatic duct epithelium express, but do not overexpress, MARS1. However, because acinar tissue are glands, they have higher MARS1 expression than normal pancreatic-duct epithelium. PDAC originates from the pancreatic-duct epithelium and so has higher MARS1 expression than normal pancreatic-duct epithelium. The MARS1 standard for PDAC was determined based on the differences in MARS1 expression levels among tissues.

‘MARS1 is strongly expressed in acinar cells (internal control cells) in normal pancre-atic tissues, and weakly expressed in the benign pancreatic duct. This is because MARS1 expression is high in acinar tissue, which are exocrine and endocrine organ, as a result of their high level of protein production. MARS1 expression is low in mu-cin-producing epithelia such as those in the pancreatic duct, which have low protein production. We used the MARS1 expression levels in acinar cells and the benign pan-creatic duct as references. High MARS1 expression was defined as equal to or stronger than that in normal acinar cells.’

Indeed, this marker has limitations in endoscopic ultrasound-guided fine needle aspiration (EUS-guided FNA). In this study, MARS1 expression was based on that in normal tissue including acinar cells and ductal epithelium. This is because it is difficult to obtain normal tissue from samples obtained by EUS-guided FNA. We plan to evaluate MARS1 expression in samples obtained by EUS-guided FNA. We have added the above as a limitation as follows:

‘Third, the standard for MARS1 expression was that in normal tissue. However, normal tissue cannot be obtained by endoscopic ultrasound (EUS)-guided fine needle aspiration (FNA). For this marker to be used clinically in all patients with PDAC, further re-search is needed to enable its application in conjunction with EUS-guided FNA.’

Thank you for the great suggestion. The grading system would be useful in studies involving EUS-guided FNA samples. In fact, a cytology grading system is used for other carcinomas.

  1. In Part 3.2 Risk factors, numbers of high MARS1 expression are change ((HR=5.663, p=0.001 vs HR=4.532, p=0.008) and confusing. I understand they are from two difference analyze tool, but it will be easier for readers to access by clarify them in the paragraph.

Answer) The high HR value for MARS1 expression at the front is a univariable result, and the one at the back is a multivariable result. An explanation has been added to the manuscript.

‘3.3 Risk factors for overall and recurrence-free survival

‘In univariate analyses, tumor size > 3 cm (hazard ratio [HR] = 1.546, p = 0.015), TNM stage (HR = 2.052, p = 0.009), lymph node metastasis (HR = 2.310, p < 0.001), lymphovascular invasion (HR = 1.438, p = 0.041), perineural invasion (HR = 2.105, p = 0.001), and high MARS1 expression (HR = 5.663, p = 0.001, Table 2) were significantly associated with a poor OS of patients with PDAC. In the multivariate analysis, high MARS1 expression (HR = 4.532, p = 0.008), lymph node metastasis (HR = 2.047, p = 0.020), and perineural invasion (HR = 1.887, p = 0.015) were independent prognostic markers for a poor OS.

Tumor size > 3 cm (HR = 1.770, p = 0.009) and high MARS1 expression (HR = 4.857, p = 0.005) were significantly associated with the DFS of patients with PDAC in univariate analyses (Table 2). High MARS expression (HR = 4.130, p = 0.013), poor differ-entiation (HR = 1.972, p = 0.042), and positive resection margin (HR = 1.750, p = 0.036) were independent prognostic markers for a poor DFS in the multivariate analysis. The multivariate analysis identified MARS1 expression as a significant and independent prognostic factor for OS and DFS. These results suggest that MARS1 expression has potential as a prognostic factor for patients with pancreatic cancer.’

In summary, these valuable data can be revised to digging more information from them. I would suggest to accept this paper after minor modification.

Best
